# Electrically and mechanically driven rotation of polar spirals in a relaxor ferroelectric polymer

Mengfan Guo [1,2,9] ✉, Erxiang Xu[1,9], Houbing Huang [3,9], Changqing Guo [3], Hetian Chen[1], Shulin Chen [4], Shan He[1], Le Zhou[1], Jing Ma [1], Zhonghui Shen [5], Ben Xu [6], Di Yi[1], Peng Gao[7], Ce-Wen Nan [1], Neil. D. Mathur [2] ✉ & Yang Shen [1,8] ✉

Topology created by quasi-continuous spatial variations of a local polarization direction represents an exotic state of matter, but field-driven manipulation has been hitherto limited to creation and destruction. Here we report that relatively small electric or mechanical fields can drive the non-volatile rotation of polar spirals in discretized microregions of the relaxor ferroelectric polymer poly(vinylidene fluoride-*ran*-trifluoroethylene). These polar spirals arise from the asymmetric Coulomb interaction between vertically aligned helical polymer chains, and can be rotated in-plane through various angles with robust retention. Given also that our manipulation of topological order can be detected via infrared absorption, our work suggests a new direction for the application of complex materials.

Flux-closure structures[1,2], vortices/antivortices[3–7], skyrmions[8–10], and merons[11,12] in oxides, metals and polymers represent nontrivial topologies in which a local polar/magnetic order undergoes quasi-continuous spatial variations in a host crystal lattice. These structures are now extensively studied due to emergent functionalities[13–15], but the application of electrical/mechanical fields has so far only served to destroy the polar topologies of interest[16–19].

Here, we modify a ferroelectric polymer, poly(vinylidene-fluoride-*ran*-trifluoroethylene) [P(VDF-TrFE)], in which toroidal polar topology can be thermally switched[20,21]. We increase the TrFE monomer ratio to induce conformational disorder[22–24] and noncollinear dipoles[25,26], resulting in relaxor behavior. Enhanced local toroidal order with spiral topology is observed within discretized microregions smaller than 1

μm, reminiscent of the classic polar nanoregion model for relaxor ferroelectrics[27]. Continuous non-volatile rotation of these in-plane polar spirals can be achieved by applying out-of-plane electric fields or mechanical stresses, cf. ferroelectric domain switching. Asymmetric interactions and weak anisotropy of helical chain conformations facilitate the Néel rotation[28] of interchain dipoles that stabilizes the polar spirals. The rotated polar spiral can persist for at least 25 weeks with few changes, outperforming switched domains with depolarization[29]. The polar spirals can absorb polarized infrared radiation in a way that depends upon the rotation, enabling optical read-out. Our observations should inform the design of flexible materials and relaxor ferroelectrics with complex topologies, and offer guidance for field-controllable manipulations of emergent polar topologies.

[1]State Key Lab of New Ceramics and Fine Processing, School of Materials Science and Engineering, Tsinghua University, 100084 Beijing, China. [2]Department of Materials Science, University of Cambridge, 27 Charles Babbage Road, CB3 0FS Cambridge, UK. [3]School of Materials Science and Engineering & Advanced Research Institute of Multidisciplinary Science; Beijing Institute of Technology, 100081 Beijing, China. [4]Changsha Semiconductor Technology and Application Innovation Research Institute, College of Semiconductors (College of Integrated Circuits), Hunan University, 410082 Changsha, China. [5]International School of Materials Science and Engineering, Wuhan University of Technology, 430070 Wuhan, China. [6]Department of Graduate School, China Academy of Engineering Physics, 100193 Beijing, China. [7]Electron Microscopy Laboratory and International Center for Quantum Materials, School of Physics, Peking University, 100871 Beijing, China. [8]Center for Flexible Electronics Technology, Tsinghua University, 100084 Beijing, China. [9]These authors contributed equally: Mengfan Guo, Erxiang Xu, Houbing Huang. ✉e-mail: mg2129@cam.ac.uk; ndm12@cam.ac.uk; shyang_mse@mail.tsinghua.edu.cn

## Results

### Observation of polar spirals

By melt-recrystallizing spin-coated thin films of P(VDF-TrFE) (see dielectric properties in Supplementary Figs. 1 and 2), a 100-nm-thick monolayer of face-on lamellae with vertically aligned polymer chains were produced (Fig. 1a and Supplementary Fig. 3a–c). The interchain dipoles arising from the electronegativity difference are distributed in the film plane[20]. We therefore used in-plane piezo-response force microscopy (IP-PFM) to characterize the film. Curly stripe domains with width of ~20 nm are predominant (Fig. 1b, c and Supplementary Fig. 3g–i), and they exhibit coiling and coarsening in randomly distributed microregions of typical size ~1 μm, leading to a series of concentric ring-shaped domains (Fig. 1b, c and Supplementary Fig. 3j–l). The curly stripe domains and the concentric ring-shaped domains both indicate a whirl of local polarization. To semiquantitatively analyze the inhomogeneous whirls, we mapped the nominal toroidal order (Fig. 1d) by calculating domain-wall curvature under the assumption that the local polarization is parallel to the nearest domain wall. A matrix with low curvatures of ~$10^{-4}$ nm$^{-1}$ (Fig. 1d) contains microregions with high curvatures of ~$10^{-2}$ nm$^{-1}$ corresponding to the concentric ring-shaped domains. This suggests a great increase of local toroidal order within confined microregions, which is analogous to the concept of polar nanoregions in relaxor ferroelectrics.

Angle-resolved IP-PFM measurements have been performed on an identical microregion (Fig. 1e, f and Supplementary Fig. 4). The IP-PFM phase images vary with respect to each other when the measurement axis is rotated from vertical (Fig. 1e) to horizontal (Fig. 1f), and a blue disk domain appears at the center. The domain walls shift outward for

counterclockwise rotations of the measurement axis (Supplementary Fig. 4a–e and Supplementary Movie 1a). A polarization map derived from the angle-resolved results appears in Supplementary Fig. 4f–h. By assuming a finite calculus limit, we calculated the curl ($\nabla \times \mathbf{P}$, Fig. 1g) and divergence ($\nabla \cdot \mathbf{P}$, Fig. 1h) of the local polarization $\mathbf{P}$ (Supplementary Fig. 4f), both of which present a double spiral that fills nearly all of the microregion. The positive/negative curl (red/blue spiral in Fig. 1g) denotes a counterclockwise/clockwise (CCW/CW) polarization rotation. The positive/negative divergence (red/blue spiral in Fig. 1h) denotes a polar source/sink. We thus infer the topology schematized in Fig. 1i, where a polar source that spirals inwards (red arrows), and a polar sink that spirals outwards (blue arrows), are separated by CW or CCW Néel rotations of polarization (white arrows). Low-angle domain walls are identified at locations where the local polarization is perpendicular to the measurement axis (Supplementary Fig. 4f, g). We choose to use the fact that the polar sink spirals outwards to classify the polar spiral in Fig. 1 as a CCW polar spiral, and the Néel rotation is CCW when moving outwards along all radial directions (Supplementary Fig. 4g, h).

As expected, we have also observed CW polar spirals, which rotate in the opposite sense (Supplementary Fig. 5). For CCW rotations of the measurement axis, domain walls shift inward (Supplementary Fig. 5a–d and Supplementary Movie 1b) as the Néel rotation is CW when moving outwards along all radial directions (Supplementary Fig. 5f, g). Both types of polar spiral can be observed in the same thin film (Supplementary Fig. 6), but the probability of observing both types of polar spiral was found to be asymmetric, as the CW/CCW ratio for 148 polar spirals in two films is 0.85. It should be noted that CW/

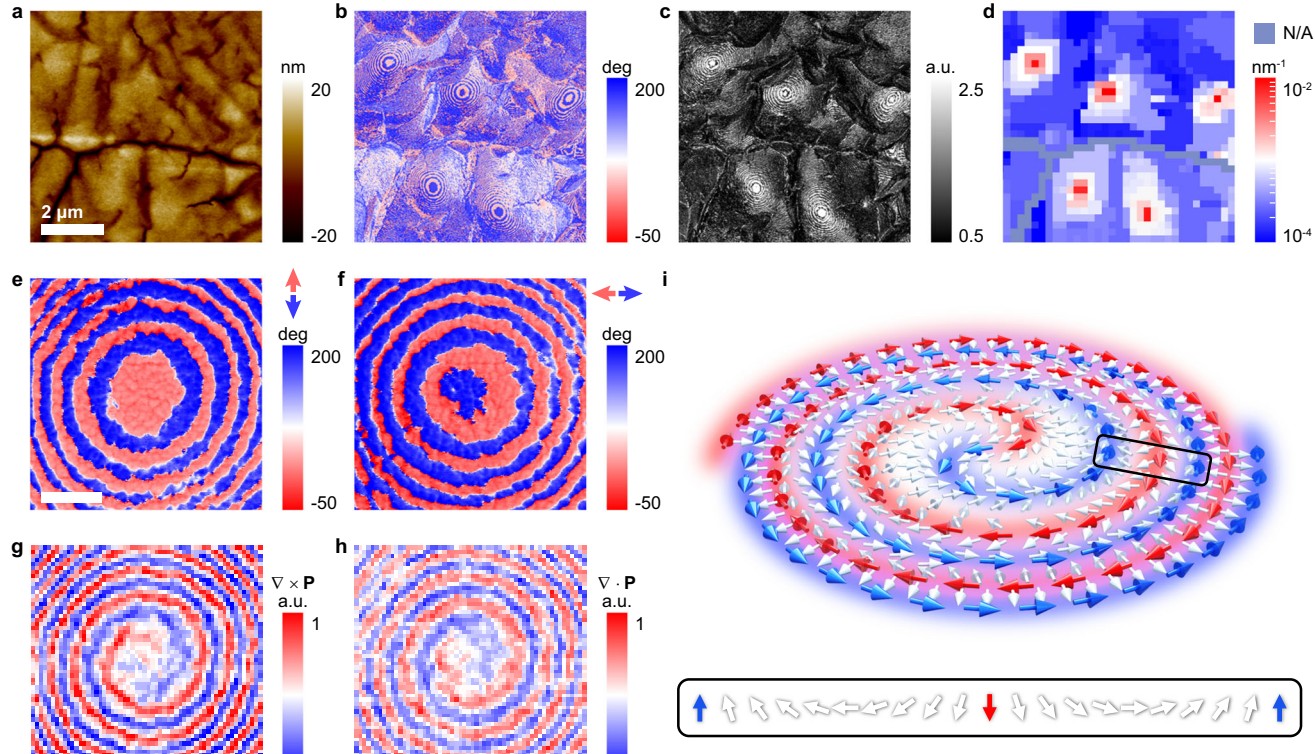

**Fig. 1 | Observation of a microregion containing an in-plane polar spiral.** **a** Morphology of a melt-recrystallized thin film of the relaxor ferroelectric polymer. The scale bar is 2 μm. IP-PFM phase (**b**) and amplitude (**c**) images of the same area in **a** exhibiting concentric ring-shaped domains in curly stripe domains. **d** Distribution of domain wall curvatures in the same area in **a–c** evidencing nominal toroidal order. It is assumed that the local polarization is parallel to the nearest domain wall so that larger curvature (denoted red) reflects stronger toroidal order. IP-PFM

phase images of identical concentric ring-shaped domains with the axis along vertical (**e**) and horizontal (**f**) measurement directions. The scale bar is 0.3 μm. The curl (**g**) and the divergence (**h**) of local polarization in the same area as **e** and **f**, revealing the polar spiral topology. **i** Schematic stereoscopic view of a CCW polar spiral, arrows represent regions of polarization. The red/blue arrows denote the polar source/sink that spirals in/out. The white arrows represent Néel rotation along the radial direction, as shown in more detail via the inset.

CCW polar spirals possess a positive/negative electric toroidal moment (Supplementary Fig. 7), indicating that the CW and CCW polar spirals might be mono-chiral and can be converted to each other by a 180° rotation along any in-plane axis. Thus, the uneven distribution might be attributed to the dielectric asymmetry between the air and the substrate.

## Rotation of polar spirals

To examine field-response of the polar spirals, we applied electric bias, or mechanical pressure through scanning tips to the microregions. Continuous and non-volatile rotations of the polar spirals were observed after applying either electric or mechanical fields, as illustrated in Fig. 2a. Domain walls of microregions in IP-PFM images have been observed to collectively shift inward, for either negative or positive bias applied through conducting tips (Supplementary Fig. 8 and Supplementary Movie 2). According to the polarization maps, the spiral geometry remains the same after electrically driven changes, except that it is rotated around the center (insets in Fig. 2b and Supplementary Fig. 8). We therefore identify the change as a rotation of the polar spirals, and define CW rotation to have a positive sign. The changes in the rotational angle of the CW and CCW polar spirals versus the applied tip bias are plotted in Fig. 2b. When the bias of ~−6 V is exceeded in magnitude (electric field of <−60 MV m$^{-1}$), the rotation becomes evident. This rotation persists after withdrawing the bias, and rotation increases with a greater voltage (Fig. 2b). The CW polar spiral is rotated in the CW direction (upper half of Fig. 2b), and symmetrically, the CCW polar spiral is rotated along the CCW direction (lower half of Fig. 2b). The rotational direction is the same when the bias takes a positive sign (Supplementary Fig. 8g–l and Supplementary Fig. 9), suggesting that the electrical rotation should be attributed to the electrostrictive effects which are coupled to the square of electric field. When the magnitude of the tip bias exceeds ~20 V (electric field >200 MV m$^{-1}$), the polar spiral transform into a monodomain (Supplementary Fig. 10g, h).

Similar rotational changes can also be induced by mechanical stresses applied through scanning tips. Domain walls in IP-PFM images have also been observed to collectively shift inward (Supplementary Fig. 11 and Supplementary Movie 3). When forces exceed ~8 nN (compressive stresses >214 MPa, see Methods), a CW/CCW polar spiral starts to rotate along the CW/CCW direction (Fig. 2c and Supplementary Fig. 11). It transitions into a series of stripe domains when the force exceeds ~46 nN (stress >383 MPa). In our studies, the electrically induced rotation can reach 130°, while the mechanically induced

rotation can exceed 780°. The narrower range of rotation induced by electric fields should be attributed to the indirect manipulation through electrostriction, and the early destruction of polar spirals induced by the plastic deformation and the large local electric field near the biased tip[30] that exceeds the switching coercive field. Although the sense of rotation is independent on the external field, we can in effect reset a polar spiral back to its initial state after rotating it by 360° (Supplementary Fig. 12). Besides, the sense of polar spirals (CW or CCW) is strongly protected against the electric or mechanical field whilst manipulation.

In addition to the electrical and mechanical stimuli from the tip, we addressed entire films (on substrates) via corona poling or uniaxial pressure (Supplementary Fig. 10a–f), and inward domain-wall shifting is still observed. While the spiral topology is preserved when electric field and mechanic stress are below ~500 MV m$^{-1}$ and ~390 MPa, respectively, it transforms into either a uniform monodomain or a series of stripe domains in larger fields similar to tip-induced transitions (Supplementary Fig. 10g–n). By comparing the polarization maps before and after applying fields (Supplementary Fig. 13), each local polarization in the polar spiral exhibits rotation along the same direction by almost the same angle. This collective rotation of local polarizations leads to the rotation of polar spirals (Supplementary Movies 4, 5). Thus, the rotation is significantly different from classic ferroelectric domain switching.

## Formation of polar spirals

We attribute the emergence of polar spirals and their persistence in external fields to the stereoirregular chain arrangement. Both the main scattering arc at the $Q_y$ axis and the diffraction points (Supplementary Fig. 14a, b) correspond to the (110) and (200) planes, which have very similar spacings in the pseudo-hexagonal lattice, suggesting the face-on lamellae are single crystals with polymer chains ($c$ axis) aligned along the film normal. Nevertheless, splitting of the (110)/(200) peak can be observed after integrating the grazing incidence wide angle X-ray scattering (GI-WAXS) intensity (Fig. 3a). We therefore fitted the experimental data (black line) to two Gaussian peaks (cyan and pink area) centered at $Q = 13.49$ nm$^{-1}$ and $Q = 13.15$ nm$^{-1}$, with the fitted intensity (red line) agreeing well with the experiment. Similar splitting has also been observed in electron diffraction (Supplementary Fig. 14c), which should be attributed to the coexistence of *trans*-planar and 3/1-helical phases[22,25,26]. Indeed, in the Fourier-transform infrared (FTIR) spectrum (Supplementary Fig. 14d), we observed characteristic bands corresponding to all-

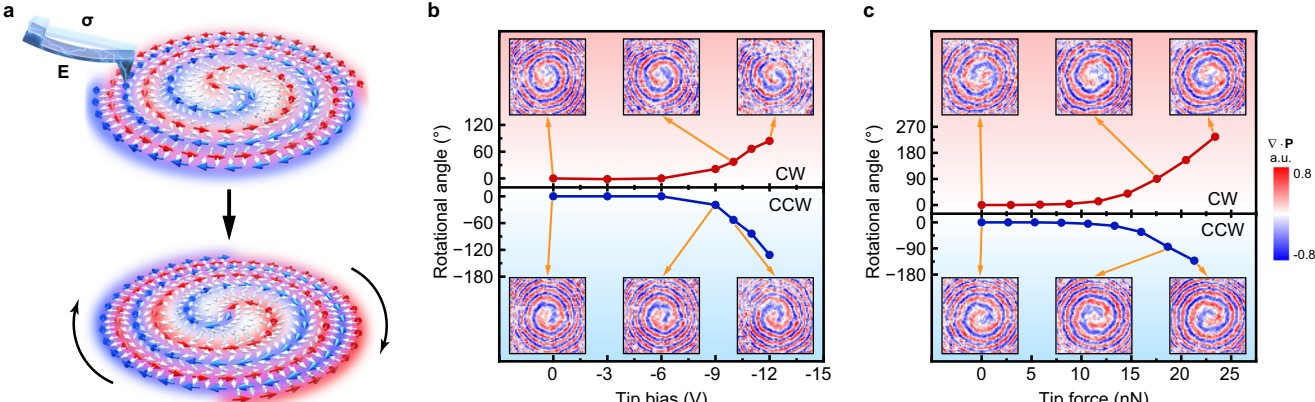

**Fig. 2 | Field-induced rotation of polar spirals. a** Schematic illustration showing field-manipulated rotational change of a polar spiral. After the application of either electric field (**E**) or mechanical stress (**σ**), a CW (CCW) polar spiral will rotate along CW (CCW) direction. **b** Electric-field-induced rotation of a CW polar spiral (red data) and a CCW polar spiral (blue data). For selected data points, the insets in red (blue) panel show the CW (CCW) polar spiral rotated along the CW (CCW) direction. **c** Stress-induced rotation of a CW polar spiral (red data) and a CCW polar spiral (blue data). For selected data points, the insets in red (blue) panel are the CW (CCW) polar spiral rotating along the CW (CCW) direction. All insets in **b** and **c** show the divergence of local polarization, and the side length of each inset is 1.138 μm.

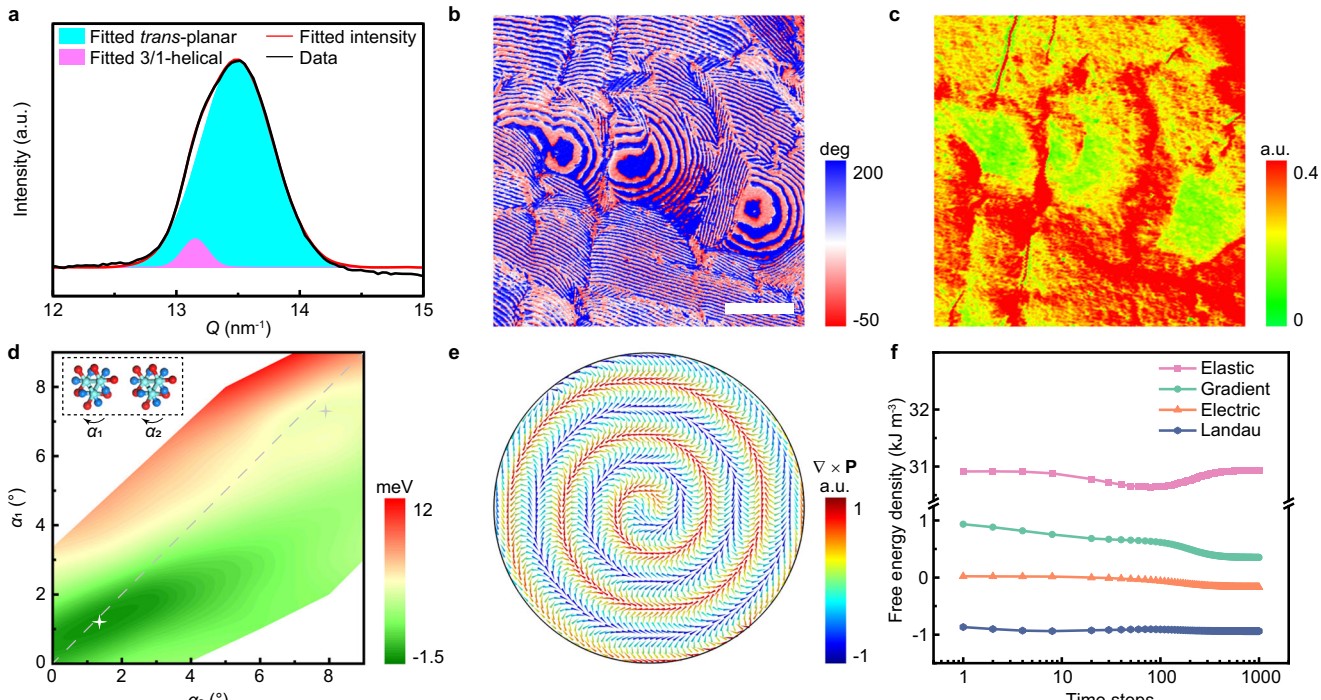

**Fig. 3 | Structural characterizations and calculations for the polar spirals.**
**a** Integrated GI-WAXS intensity near the main scattering arc (black line, Supplementary Fig. 14a) and a fit of the intensity profile to two Gaussian peaks. The cyan and pink areas denote the fitted peak, corresponding to *trans*-planar and 3/1-helical phases, respectively. The fitted intensity (red line) agrees well with the experimental one. IP-PFM phase image (**b**) and out-of-plane AFM-IR absorption image at 880 cm$^{-1}$ (**c**) of the same area. Microregions exhibit decreased absorption intensity compared with the matrix. The scale bar is 0.8 μm. **d** A map of gain in free energy per chain for a pair of (TG)$_3$ helical chains with each chain individually rotated by $\alpha_1$ and $\alpha_2$, compared with the state of $(\alpha_1, \alpha_2) = (0°, 0°)$. Both the global [white star, $(\alpha_1,$ $\alpha_2) = (1.2°, 1.35°)$] and local [gray star, $(\alpha_1, \alpha_2) = (7.6°, 7.9°)$] lowest energy states are located at the same noncollinear region with $\alpha_1 < \alpha_2$. The inset shows the sketch of two (TG)$_3$ helical chains and how they are rotated. The red, blue, and cyan atoms are F, H, and C, respectively. **e** Polarization vector map from a phase-field simulation of a thin film with low anisotropy, predicting a CCW double-spiral polar texture similar to the experimental observations. The color of vectors depicts the divergence of local polarization ($\nabla \cdot \mathbf{P}$). **f** Time-dependent evolution of elastic (pink), gradient (green), electric (orange), and Landau (blue) energies from symbol simulations on electric-induced rotation of a polar spiral.

*trans* conformation at 1287 and 888 cm$^{-1}$, and to the 3/1-helical conformation at 508 cm$^{-1}$. We calculated the strain-free lattice spacings as a reference according to the X-ray diffraction (XRD) profile of edge-on thick films[31] (Supplementary Fig. 14e), and we compare them with values from the GI-WAXS results (Supplementary Table 1). There is a little strain in the thin film for both phases as indicated by the nearly identical measured spacings compared to the reference value. It is therefore suggested that compared with the strained non-morphotropic polymer[20], the macroscopic strain plays a less important role here in the formation of toroidal order and its influence is negligible in the following discussion.

The microscopic phase coexistence is confirmed by PFM and atomic force microscope infrared spectroscopy (AFM-IR) measurements (Fig. 3b, c). Three isolated microregions are shown in Fig. 3b (also see Supplementary Fig. 14f–h). When the same microregion is subjected to AFM-IR measurement using an IR beam polarized along out-of-plane (OOP) direction near the characteristic bands of the *trans*-planar phase, it exhibits a quenched IR absorption compared with the matrix (Fig. 3c and Supplementary Fig. 14i–k). We therefore suggest the microregions mainly consist of the 3/1-helical phase, whereas the matrix is dominated by the *trans*-planar phase. The 3/1-helical phase contains (TG)$_3$ or (T$\bar{\text{G}}$)$_3$ conformation (T for *trans* and G for *gauche*), with either one generating chains with one sense of helicity periodically in three -CH$_2$-CF$_2$- segments (Supplementary Fig. 15a), in line with the Néel rotation along different directions observed in CW and CCW polar spirals.

We established pairs of collinear and noncollinear chains with the (TG)$_3$ conformation to compare the difference in free energies

(Fig. 3d). Two neighboring helical chains along the [010] direction were selected. Their initial alignment is adopted from the reported prediction[22], which is in the vicinity of the collinear alignment with the largest energy gain (Supplementary Fig. 15b). The two helical chains, denoted in the inset of Fig. 3d, are then rotated in CW direction by $\alpha_1$ and $\alpha_2$, respectively. The gains in free energy per chain compared with the state of $(\alpha_1, \alpha_2) = (0°, 0°)$ have been calculated and are shown in Fig. 3d.

The gray dashed line in Fig. 3d is a collection of points with $\alpha_1 = \alpha_2$, denoting the collinear states. The region beneath the line represents noncollinear states with $\alpha_1 < \alpha_2$, whereas the region above represents $\alpha_1 > \alpha_2$. We found three striking features in Fig. 3d: 1) The most stable polarization state with the largest free energy gain is located in the $\alpha_1 < \alpha_2$ zone, at $(\alpha_1, \alpha_2) = (1.2°, 1.35°)$ (the white star); 2) The polarization state with the local lowest energy appears at $(\alpha_1, \alpha_2) = (7.3°, 7.9°)$ (the gray star), again lying in the $\alpha_1 < \alpha_2$ zone; 3) The zone of $\alpha_1 > \alpha_2$ exhibits a more dramatic rise in free energy (red color at the upper edge) than the $\alpha_1 < \alpha_2$ zone (green color at the lower edge). These features confirm that helical chains tend to self-organize into noncollinear polarization states, whereas chains with the (TG)$_3$ conformation tend to result in Néel rotation with $\alpha_1 < \alpha_2$, and symmetrically, (T$\bar{\text{G}}$)$_3$ chains should result in Néel rotation with $\alpha_1 > \alpha_2$. Tilted intermolecular interactions have been found between neighboring chains (Supplementary Fig. 15c, d), which should facilitate the inhomogeneous rotation. Continuous spatial rotation of polarization also requires low dielectric anisotropy. We therefore performed phase-field simulations with strong and weak anisotropy (Supplementary Fig. 16 and Supplementary Tables 2 and 3). The low depth of the polarization energy surface should enable easier

spatial rotation[32]. Thus, the simulated polarization map in Fig. 3e with weaker anisotropy indicates a polar spiral resembling our experimental observations.

Time-dependent phase-field simulations in electric fields have also been performed. Polar spirals have indeed been predicted to rotate, and would be broken into smaller polar clusters in large fields (Supplementary Figs. 17 and 18). Energy components in 1,000 time steps are collected after the field application (Fig. 3f). Before time steps of 100, the elastic energy (pink) quickly decreases to overcome a small energy barrier and there is a rise in the Landau energy (blue), suggesting an electromechanical conversion. The slight reduction in gradient energy (green) indicates that local toroidal order is suppressed, which may be in favor of dynamic rotational changes. The electric energy (orange) remains nearly unchanged, due to the conserved spiral skeleton in the polar spiral. After time steps of 100, the elastic energy is restored at the expense of decreased electric and gradient energies, suggesting a cross-coupling of electromechanical energy conversion with toroidal order. Thus, electromechanical interconversion triggered by the electric fields involves the evolution of local polarizations, leading to the dynamic rotational change within the polar spirals. A similar energy evolution process has been found in the stress-induced rotation (Supplementary Fig. 19).

### Retention of field-manipulated polar spirals

The retention of a rotated polar spiral has been examined and compared with a switched domain (Fig. 4). The rotated polar spiral persisted with nearly no change after 25 weeks (upper insets of Fig. 4), such that the calculated core area changed by ≤2%. The domain in a microregion was switched by addressing it via a biased PFM tip that was rastered over the entire domain, and exhibits fractured switched-back domains right after the switching (blue stripe in the lower left inset of Fig. 4). With prolonged retention time, more fractured domains in blue were switched back (lower insets of Fig. 4). By calculating the unswitched area of the central region, we found the retention ratio of the deterministically switched region dropped to 86% after 25 weeks. The continuous spatial rotation in the polar spirals avoids any sharp discontinuities of polarization, and hence largely suppresses any depolarization fields that could lead to the nucleation and growth of anti-domains.

### Optical read out of field-manipulated polar spirals

AFM-IR measurements have been conducted on a microregion to investigate its optical response after field-manipulation, and the domain states (Fig. 5a–c) are consistent with the IR absorption patterns (Fig. 5d–f). Figure 5a shows the original domain states of a microregion. By projecting an IR beam at 1400 cm⁻¹ polarized along IP direction, AFM-IR absorption of the microregion exhibits concentric feature with a periodically varying intensity along the radial direction (Fig. 5d), as predicted by simulation (Supplementary Fig. 20). The IR absorption is also weaker (purple in Fig. 5d) in blue-domain regions and stronger (orange in Fig. 5d) in red-domain regions. After rotating the polar spiral CW by 86°, the collective inward domain-wall shifting (Fig. 5b) leads to the same shifting of the absorption pattern (Fig. 5e). The elliptical weaker absorption region at the center of Fig. 5d disappears, and instead the orange central region in Fig. 5e corresponds to stronger absorption. After further rotating the polar spiral CW by another 88°, the domain state and absorption pattern display simultaneous inward shifting (Fig. 5c, f).

In summary, we report the observation of polar spirals within toroidal polar microregions in thin films of relaxor ferroelectric polymers. This represents an unequivocal observation of ordered polar topology in systems that are considered to be "hopeless messes"[33]. The polar spirals can be rotated by applying electric fields or mechanical stresses, enabling non-destructive, non-volatile, and continuous rotations, which may have influence on the macroscopic dielectric properties and is promising for multistate memories and neuromorphic systems[34]. The formation and stabilization of these polar spirals are induced by the asymmetric dielectric interaction and weak dielectric anisotropy within the 3/1-helical phase. The rotated topologies display excellent retention, and infrared absorption pattern permits optical read out, e.g. for use in field-controllable spatial light modulators and structured light beams[35,36]. Our observation and manipulation of polar spirals should provide insights and design principles for integrating field-controllable topology into relaxor ferroelectrics.

## Methods
### Thin film fabrication
The copolymer poly(vinylidene fluoride-*ran*-trifluoroethylene) [P(VDF-TrFE)] 50/50 mol% was obtained from Arkema Piezotech and used as-received. The copolymer powder was dissolved into methyl ethyl ketone at concentration of 2% w/v. The films were firstly spin-coated onto substrates at 5000 rpm. The substrates include silicon wafer with

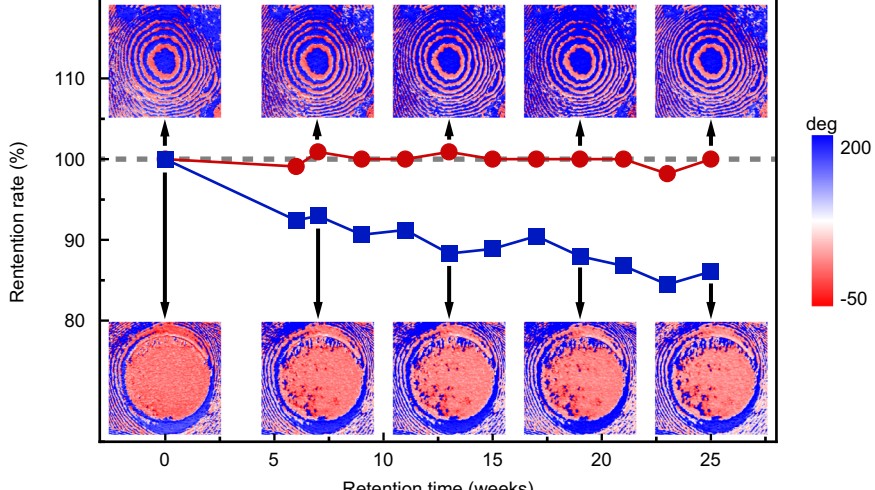

**Fig. 4 | Retention behavior of polar spirals after field-manipulation.** Retention of a continuously rotated polar spiral (red data) and a switched domain (blue data). The insets show the IP-PFM images of the polar spiral at various times after the field-manipulated rotation (images over the red data) and switching (images under the blue data). The side length of each inset is 0.981 μm.

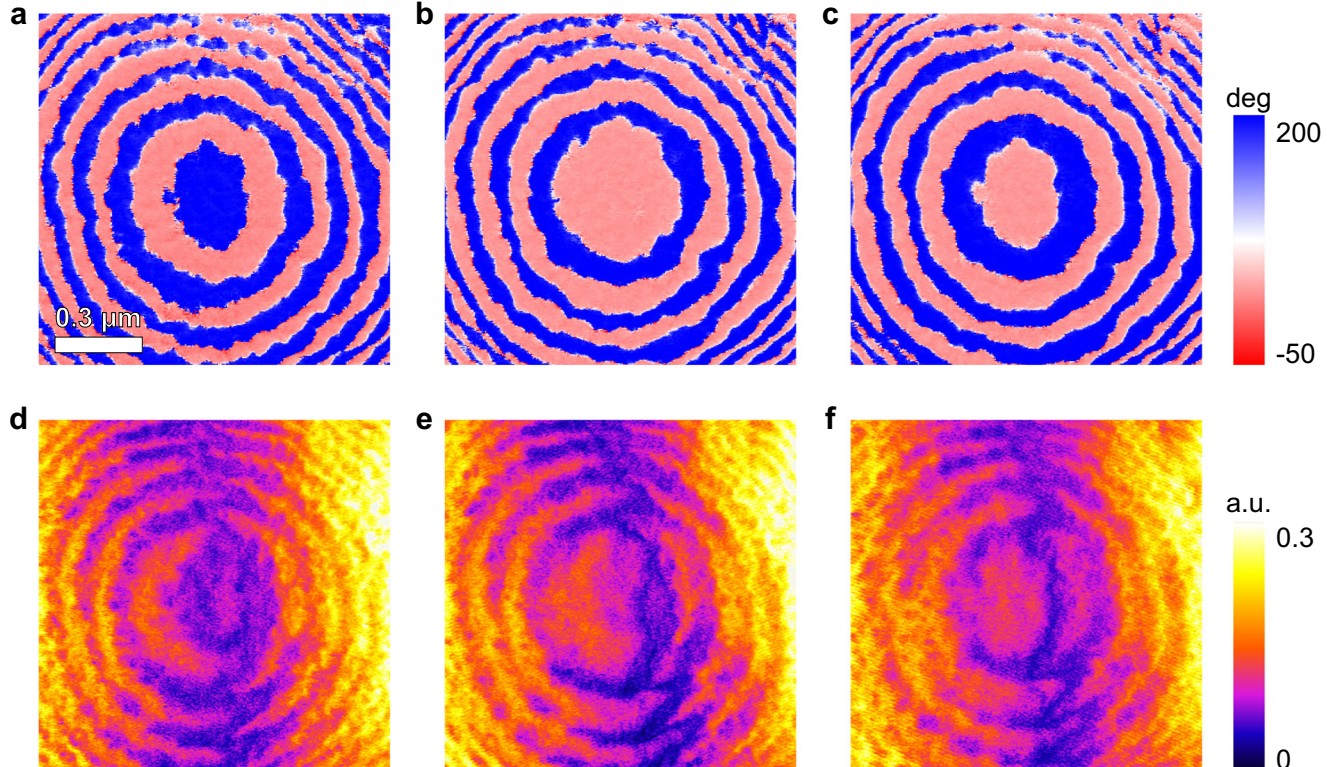

**Fig. 5 | Optical response of a polar spiral after field-manipulation.** IP-PFM images of a polar spiral before any rotation (**a**), after CW rotation by 86° (**b**) and further CW rotation by 88° (**c**). The scale bar is 0.3 μm. AFM-IR images with the IR beam at 1400 cm⁻¹ polarized along IP direction of the same region as **a**–**c** before any rotation (**d**), after CW rotation by 86° (**e**) and further CW rotation by 88° (**f**).

platinum top layer, quartz wafer with gold top layer, and quartz wafer. After dried in a vacuum oven at 30 °C for 1 h, the films were annealed at 240 °C for 10 min, and the cooling rate was controlled around 1 °C min⁻¹. The film thickness was around 100 nm confirmed by atomic force microscope (AFM).

**Scanning probe microscopy measurements**

A commercial scanning probe microscope (Asylum, MFP-3D infinity) was applied for AFM and piezo-response force microscope (PFM) measurements. In AFM, in-plane (IP) PFM tests, and PFM lithography tests, Si cantilevers with conducting Pt/Ir coating layer of spring constant around 0.2 N m⁻¹ and tip radius around 25 nm (NanoWorld, Arrow-CONTPt) were used. For PFM imaging, Vector mode was used where tips were modulated at around 240 kHz for IP imaging, with the AC voltage set at 2 V. The images obtained by Vector Mode were double checked by using dual AC resonance tracking (DART) mode and the patterns can be reproduced. For angle-resolved IP-PFM tests, the rotation of sample was controlled by a protractor. To ensure identical position was imaged after rotating the sample, we made cross-scratches as a mark on the sample surface in advance. This method was applied to locate the scanning position in other situations if we had to move the sample in between the scanning probe microscopic studies.

For electric-field-induced manipulations using PFM lithography, the DC voltage on tip was previously edited in the software. The scan speed was set at 1.95 Hz and no AC voltage was applied during the scanning. The DC voltage was divided by film thickness (100 nm) to obtain the electric field value. And an electric field with downward direction is defined with a positive sign.

For stress-induced manipulations, the deflection value of the PFM cantilever, which is a signal from photodetector, was preset to control the stress/force applied onto the sample. The difference in deflection value between a pressed cantilever and a free cantilever reflects how hard the tip and sample surface are pressed to each other. To obtain the force value $F$, we first calibrated the tips by the thermal noise method, and obtain the inverse optical lever sensitivity (InvOLS) and the spring constant $k$ of the tips[37]. The force could be obtained by:

$$F = \mathrm{InvOLS} \times k \times \Delta\mathrm{Deflection} \qquad (1)$$

We utilized Hertzian contact mechanics of a spherical indenter to simulate the stress field of the film under an AFM tip[38]. For simplicity, the stress $\sigma_{33}$ at the contact center was used to represent the stress distribution under certain loading force, and it can be expressed as:

$$\sigma_{33} = -\frac{1}{\pi}\left(\frac{6FE_e^2}{R^2}\right)^{\frac{1}{3}}, \qquad (2)$$

where $F$ is the AFM tip loading force, $R$ is the radius of the AFM tip (~25 nm), and $E_e$ is the effective modulus. The $E_e$ is defined by:

$$\frac{1}{E_e} = \frac{1-\nu_s^2}{E_s} + \frac{1-\nu_t^2}{E_t} \approx \frac{1-\nu_s^2}{E_s}, \qquad (3)$$

where $\nu_s$ is the Poisson ratio of the sample, $E_s$ is the Young's modulus of the sample, $\nu_t$ is the Poisson ratio of the tip, and $E_t$ is the Young's modulus of the tip. While $\nu_s$ and $\nu_t$ are similar in value, since $E_t$ is two orders of magnitude larger than $E_s$, the effective modulus $E_e$ can be approximately calculated from $\nu_s$ and $E_s$. The value of $\nu_s$ is 0.305 (ref. 39), and the value of $E_s$ (1.8 GPa) was obtained from a nanoindentation experiment.

For retention measurement of field-manipulation, the retention rate of a rotated polar spiral was calculated by comparing the area

encircled by the seventh smallest circular domain wall (upper insets in Fig. 4). The retention rate of a switched microregion was calculated by comparing the non-switching-back area within the lithographed region (lower insets in Fig. 4).

A commercial scanning probe microscope equipped with a quantum cascade laser (Anasys, NanoIR2) was used for AFM-IR measurements. The tips customized and provided by Anasys were contact-mode tips with spring constant between 0.07 to 0.4 N m$^{-1}$. By operating in the software, the laser can be modulated between 750 cm$^{-1}$ to 1900 cm$^{-1}$ and be polarized along OOP or IP directions.

The polar spiral regions we characterized and manipulated are smooth in surface without cracks (surface roughness $R_a = 1.568$ nm for Supplementary Fig. 3j).

## Structural characterization

Optical microscopy measurements were completed by using Olympus BX53M in the reflection mode.

GI-WAXS measurements were conducted at beamline BL16B1 of Shanghai Synchrotron Radiation Facility. X-rays with photon energy of 10 keV were incident on the thin film sample in a rough vacuum environment ($10^{-3}$ Torr range) at incident angles of 0.10° and 0.14°, below the critical angle of organic material or the Si substrate, for a 5-second exposure. A pixel array detector (Pilatus 2M, Dectris) was positioned 228 mm from the sample. The spacing $d$ of any scattering unit can be calculated as

$$d = 4\pi/Q, \tag{4}$$

where $Q$ is the reciprocal spacing.

The TEM experiments were carried out at FEI Tecnai F20 at 200 kV by a liquid nitrogen side-entry specimen holder (Gatan 636). The dose rate is estimated to be 0.25 e Å$^{-2}$ s$^{-1}$ during the selected area electron diffraction (SAED) tests. Silicon wafers with silicon oxide layer whose thickness was 300 nm were used as substrate for film deposition. By chemically etching of the substrate in solution with pH value of 14, a layer of free-standing face-on lamellae was obtained. After washed by deionized water, the film was floated onto copper grids. The aperture diameter is 3.8 μm for obtaining SAED pattern.

FT-IR was carried out on a Nicolet iS50 spectrometer (Thermo Fisher) under grazing incidence mode. The spectra were averaged from 128 scans.

X-ray diffraction (XRD) $\theta$-$2\theta$ scanning measurements were completed by using PANalytical Empyrean with Cu Kα radiation. The calculation of lattice spacing $d$ follows by the Bragg law, also written as

$$2d \sin \theta = \lambda_{Cu-K\alpha}, \tag{5}$$

where $\lambda_{Cu-K\alpha}$ is the wavelength of diffraction source.

## First-principle computations

First-principle computations were implemented using the Vienna Ab initio Simulation Package. The projector augmented-wave approach was used for the treatment of the core electrons. The generalized gradient approximation using the Perdew-Burke-Ernzerhof functional was employed to approximate the electron exchange correlation interactions. For simplification, PVDF chain was taken for modeling. For the calculations on free energies of (TG)$_3$ helical chain pairs, the chains are set with the distance of 0.55 nm in vacuum and get individually rotated around the geometric center to yield a series of collinear and noncollinear states. At least 20 Å of vacuum in the $x$ and $y$ directions was used to separate the chain pairs in order to avoid the artificial interaction among the periodic units. The charge density difference is obtained by subtracting the charge density of the two individual PVDF chains from the charge density of helical chain pairs with the most

stable angle. A cutoff energy 400 eV was applied for all works. The Brillouin zone was sampled by a Monkhorst–Pack $1 \times 1 \times 5$ $k$-point grid.

## Phase-field simulations

In phase-field simulations, the ferroelectric domain structures in P(VDF-TrFE) films are described by polarization vector $\mathbf{P} = (P_x, P_y, P_z)$. The temporal evolution of the polarization field is solved by the time-dependent Ginzburg–Landau (TDGL) equation, which takes into account the coupling and competition among various energies in the system, leading to the minimization of the total energy:

$$\frac{\partial P_i(\mathbf{r},t)}{\partial t} = -L \frac{\delta F}{\delta P_i(\mathbf{r},t)}, i = x,y,z, \tag{6}$$

Here, $\mathbf{r}$ is the spatial coordinate, $t$ is the evolution time, $L$ is the kinetic coefficient that is related to the domain evolution, and $F$ is the total free energy that includes the contributions from the Landau energy, the gradient energy, the elastic energy, and the electric energy:

$$F = \iiint_V (f_{Land}(P_i) + f_{grad}(P_{i,j}) + f_{elas}(P_i, \varepsilon_{ij}) + f_{ele}(P_i, E_i)) dV. \tag{7}$$

The Landau energy density $f_{Land}$ is given by,

$$f_{Land} = \alpha_{ij} P_i P_j + \alpha_{ijkl} P_i P_j P_k P_l + \alpha_{ijklmn} P_i P_j P_k P_l P_m P_n, \tag{8}$$

where $\alpha_{ij}$, $\alpha_{ijkl}$, $\alpha_{ijklmn}$ are the Landau energy coefficients.

The gradient energy density $f_{grad}$ can be described as follows,

$$f_{grad} = \frac{1}{2} G_{ijkl} P_{i,j} P_{k,l}, \tag{9}$$

where $G_{ijkl}$ are the gradient energy coefficients, and $P_{i,j} = \partial P_i/x_j$.

The elastic energy density $f_{elas}$ can be written as

$$f_{elas} = \frac{1}{2} C_{ijkl} e_{ij} e_{kl} = \frac{1}{2} C_{ijkl} (\varepsilon_{ij} - \varepsilon_{ij}^0)(\varepsilon_{kl} - \varepsilon_{kl}^0), \tag{10}$$

where $C_{ijkl}$ is the elastic stiffness tensor, $e_{ij}$ is the elastic strain, $\varepsilon_{ij}$ and $\varepsilon_{ij}^0$ are the total local strain and eigenstrain, respectively. And $\varepsilon_{ij}^0 = Q_{ijkl} P_k P_l$, where $Q_{ijkl}$ are the electrostrictive coefficients. In this study, only the linear elastic region is considered when simulating the low-field response of the polar spirals. Thus, the stresses applied in simulation do not exceed 12.0 MPa, which is lower than the assumed elastic constant by 2–3 orders of magnitude (Supplementary Table 3).

The electric energy density $f_{ele}$ is expressed as,

$$f_{ele} = -E_i \left( P_i + \frac{1}{2} \varepsilon_0 \kappa_{ij} E_j \right), \tag{11}$$

where $E_i$ is the electric field component, $\varepsilon_0$ is the vacuum permittivity, and $\kappa_{ij}$ is the dielectric constant.

The thickness and radius of simulated P(VDF-TrFE) nanodisk were 10 nm and 345 nm, respectively. The short circuit condition was applied in the simulation, and the temperature was set to be 25 °C. We used the finite element method to solve the TDGL equations, and the values of the parameters in the simulation are listed in Supplementary Tables 2 and 3.

## Polarization analysis based on PFM measurements

To obtain the nominal toroidal order evaluated by the local curvature, the obtained IP-PFM amplitude image was firstly divided into $33 \times 33$ arrays, and each region was then subjected to a recognition of potential domain walls and measurement of an averaged curvature radius.

To obtain polarization maps, angle-resolved IP-PFM images were first aligned to correct spatial distortion in nanoscale measurement. Positions with specific morphological characteristics were selected as reference points to determine the coordinate. After the correction, improved angle-resolved IP-PFM phase images would be divided into $64 \times 64$ arrays for deriving polarization maps. We did not use the data from a single pixel to improve statics. Each region in the arrays was assigned with a local orientation in a range 180° according to the direction of measurement axis. By deducing the serial angle-resolved images on a same region but with rotated measurement axes, we refined the orientation of each point within a range of 45°, and defined its orientation to be along the angular bisector.

The curl, divergence and Laplace operator of the polarization vector field **P** were further calculated to illustrate the feature of polarization distribution in toroidal polar microregions. The curl vector describes the $z$ component of $\nabla \times \mathbf{P}$, also written as:

$$\nabla \times \mathbf{P} = (\partial P_y / \partial x - \partial P_x / \partial y)\hat{\mathbf{z}} \tag{12}$$

where $z$ is the direction of film normal, and $x$-$y$ plane describes the plane of PFM images.

Likewise, the divergence and the Laplace operator of **P** in $x$-$y$ plane can be written as:

$$\nabla \cdot \mathbf{P} = \partial P_x / \partial x + \partial P_y / \partial y, \tag{13}$$

$$\Delta \mathbf{P} = (\partial^2 P_x / \partial x^2 + \partial^2 P_x / \partial y^2)\hat{\mathbf{x}} + (\partial^2 P_y / \partial x^2 + \partial^2 P_y / \partial y^2)\hat{\mathbf{y}}. \tag{14}$$

The partial derivatives of **P** mentioned above were calculated by a weighted least-squares fitting method. More specifically, the polarization component of the target point and its six surrounding points were fitted by first-order polynomial, given the Gaussian function $f_W(d)$ as the weighting function:

$$f_W(d) = \frac{1}{\sqrt{2\pi}\sigma} \exp\left\{-\frac{[(d/d_0)\sigma]^2}{2\sigma^2}\right\} = \exp\left(-\frac{d^2}{2d_0^2}\right) \tag{15}$$

where $\sigma$ is set to be $1/\sqrt{2\pi}$, $d$ is the distance to the target point, and $d_0$ is the distance between two neighboring points.

To ensure that partial derivatives at every point are fitted with seven points, the distributions of $\nabla \times \mathbf{P}$ and $\nabla \cdot \mathbf{P}$ are given in $58 \times 58$ arrays, the distributions of the $x$, $y$ components of $\Delta \mathbf{P}$ are given in $52 \times 52$ arrays. As a result, the $\nabla \times \mathbf{P}$ and $\nabla \cdot \mathbf{P}$ exhibit double-spiral features (Supplementary Figs. 4 and 5), and the $x$, $y$ components of $\Delta \mathbf{P}$ display concentric ring-shaped features. The geometric center point $(x_c, y_c)$ of the polar spiral was defined by the Laplace operator of **P**, which is given by:

$$x_c = \sum_{x,y=7}^{58} x \cdot (\Delta \mathbf{P})^2 / \sum_{x,y=7}^{58} (\Delta \mathbf{P})^2 \tag{16}$$

$$y_c = \sum_{x,y=7}^{58} y \cdot (\Delta \mathbf{P})^2 / \sum_{x,y=7}^{58} (\Delta \mathbf{P})^2 \tag{17}$$

The normalized electric toroidal moment and the normalized polarization flux of **P** are expressed as $\hat{\mathbf{r}} \times \mathbf{P}$ and $\hat{\mathbf{r}} \cdot \mathbf{P}$, respectively, where $\hat{\mathbf{r}} = \mathbf{r}/|\mathbf{r}|$ represents the unit displacement vector from the center point to the local point. Both the normalized electric toroidal moment and the normalized polarization flux reveal double-spiral features (Supplementary Figs. 4 and 5).

To determine the field-induced changes in rotational angle of the polar spiral, we first conducted angle-resolved IP-PFM measurements and fitted the area encircled by the smallest and the second smallest domain wall versus the sample rotational angles. Since the shape of the polar spiral is unchanged during field-manipulation, the yielded fitting can be used as a reference to deduce the field-induced rotational angle. Succinctly, we measured the area encircled by the domain wall corresponding to the previous fit after the field application and compared it with the fit to obtain a characteristic angle. The characteristic angle difference between each state and the initial state would be assigned as the rotational angle of the field-manipulation. Another method we have used to determine the rotational angle is to perform angle-resolved IP-PFM measurements before and after the field-induced manipulation (insets in Fig. 2b, c and Supplementary Figs. 8 and 11). The rotational angle in each local polarization (Supplementary Fig. 13e, k) was calculated by comparing the polarization maps before and after the field-manipulation. The rotational angle of the field-manipulation was defined based on Gaussian fitting (Supplementary Fig. 13f, l). We compared the rotational angles obtained from both methods and found them to agree well with each other.

The above analysis has been implemented with the aid of a self-customized Matlab package.

## Data availability
All relevant data in the main text or the supplementary materials are available from the corresponding authors upon reasonable request. Source data are provided with this paper.

## Code availability
The codes within this paper are available from the corresponding authors upon reasonable request.

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

## Acknowledgements

We thank Yuhan Liang, Yue Wang, and Fangyuan Zhu for discussions. The authors thank the beamline BL16B1 of Shanghai Synchrotron Radiation Facility for providing the beam time and discussion, and acknowledge Electron Microscopy Laboratory in Peking University for the use of electron microscopes. M.G. acknowledges support from the Shuimu Tsinghua Scholar Program and the Royal Society - Newton International Fellowship. This work was supported by Basic Science Centre Program of NSFC with the Grant No. 52388201 (C.-W.N.), the NSF of China with the Grant No. 52027817 (Y.S.), 52073155 (Y.S.), and 51972028 (H.H.), Tsinghua University Initiative Scientific Research Program with the Grant No. 20211080092 (Y.S.), and China Postdoctoral Science Foundation with the Grant No. 2022T150343 (M.G.).

## Author contributions

M.G. and Y.S. conceived and performed this study. M.G. and E.X. fabricated the films. C.G. and H.H. performed the phase-field simulations. H.C., B.X., and D.Y. performed the first-principle computations. S.C. and P.G. performed the TEM measurements. M.G., E.X., S.H., and J.M. performed the PFM measurements. M.G., E.X., L.Z., and Z.S. performed the field manipulation. M.G. and E.X. performed the AFM-IR measurements and wrote the first draft of the manuscript. Y.S., N.D.M., and C.–W.N. revised the manuscript. All authors discussed the results and edited the manuscript.

## Competing interests

The authors declare no competing interests.
