## [Peer Review File · Nature Communications]

Electrically and Mechanically Driven Rotation of Polar Spirals in a Relaxor Ferroelectric PolymerREVIEWER COMMENTS

Reviewer #1 (Remarks to the Author):

The manuscript by M. Guo et al. reported a non-volatile rotation of polar spirals in relaxor ferroelectric polymer P(VDF-TrFE) driven by either a small electric or mechanical field. This manuscript also discussed the formation of these polar spirals, arising from the asymmetric Coulomb interaction between vertically aligned helical polymer chains. These polar spirals also show robust retention and also support optical read out by AFM-IR.

I think this work presents a new possibility to manipulate the topological polar states in relaxor ferroelectric polymers. The results are important and intriguing, which can also inspire the field to explore similar phenomena in other inorganic materials. I would recommend publishing this paper in Nature Communications after addressing a few minor points:

1. I think the title of this paper can be slightly tailored since the paper only highlights the electrically driven behavior but the actual paper presents both the electrical and mechanical driven rotation.
2. I think it is helpful to comment or explain possible mechanisms that cause the differences in the effect of electrical and mechanical driven rotation. For example, electrically induced rotation can reach 130 deg while the mechanically induced rotation can exceed 780 deg.
3. I would also like to see if the authors have conducted any macroscale electrical measurements such as dielectric permittivity measurements for polar spirals having different rotation angles. If they indeed show different electrical response, these results can also enhance the paper and provide another electrical read out for the polar spirals.
4. It is a slight pity to see that the polar spirals can only rotate along one direction given their polar direction. Can the authors also comment whether it is possible to induce rotation for them to rotate along the another direction (for example, CW polar spiral rotates along CCW direction). In that way, it is possible to have a reversible switching to reset the polar spiral back to its initial state.

Reviewer #2 (Remarks to the Author):

This work reports the non-volatile rotation of polar spirals in discretized microregions of poly(vinylidene fluoride-trifluoroethylene) random copolymer caused by small electric or mechanical fields. The results, which are well supported by abundant experimental evidences, are quite interesting and may of great importance for the application of complex materials. The methods used in this work are detail and reproducible. It meets the expected standards in this field and can thus be accepted for publication in Nature Communication essentially as is or with some minor revision.

1. As can be seen from Fig. 1a, there are cracks in the film. I wonder whether the cracks influence the measurement or not. How about the roughness of the used film?
2. Moreover, the thickness of the film used in this work is about 100 nm. Considering that the lamellar thickness of the copolymer is generally around 10 nm, does the sample contain a monolayer lamella? If not, does the folding loops between the lamellae affect the related properties?

3. In Fig. 2, the CW and CCW rotations are quite clear. However, those in Fig. 1g and h seem not so clear.
4. The scale bar presented in Fig. 1e, 3b is not clear enough.

Reviewer #3 (Remarks to the Author):

Please see the attached file.

In this work, the authors presented an interesting study on the field-driven manipulation of the non-volatile rotation of polar spirals in discretized microregions of the relaxor ferroelectric polymer. The authors demonstrated that the polar spirals arise from the asymmetric Coulomb interaction between vertically aligned helical polymer chains and can be rotated in-plane through various angles with robust retention. There are, however, a few concerns and I'd like to see them addressed before the final consideration of publication.

1. The reasons why CW spirals are more pronounced in the considered system compared to CCW counterparts remain unclear. Given that this study primarily investigates the control of polar spirals, it is essential to elucidate the underlying mechanism responsible for their formation.
2. As shown in Fig. 2, the application of either electric field or mechanical stress solely alters the rotation angle of polar spirals without affecting their orientation, whether it is CW or CCW. Is it feasible to modify the rotation of polar spirals, switching them from CCW to CW or vice versa, using external fields? It is worth noting that achieving complete control over polar topological textures necessitates the ability to manage both their formation and switching among the equivalent states.
3. In the phase field model, the authors described the elastic energy density as

$$f_{elas} = \frac{1}{2} C_{ijkl} e_{ij} e_{kl}$$

where, there exist three independent elastic constants c_{11} , c_{12} and c_{44} in the Voigt's notation. However, this description of elastic energy density assumes linear elasticity and cubic symmetry, which is not suitable for ferroelectric polymer. Ferroelectric polymers typically exhibit nonlinear behavior and lack cubic symmetry. Therefore, the authors should adjust the phase field model to appropriately account for the characteristics of ferroelectric polymer. Additionally, it is important to note that the elastic energy density has a significant contribution to the total energy of the considered system, as shown in Fig. 3f. This raises a critical concern about the validation of phase field simulations due to the present assumption.

4. There is controversy surrounding the use of the electric toroidal moment in Fig. S4. As depicted in Fig. S4a and c, both CCW and CW polar spirals exhibit a distribution of the electric toroidal moment comprising both local negative and positive components. However, when these local components are summed across the entire system, their contributions nearly cancel out, resulting in a value close to zero. In this study, if I understand correctly, the authors determine the overall toroidal moment by performing path integrals. However, it is important to note that the overall toroidal moment is heavily influenced by the selected integration path. By altering the chosen paths, the magnitude and sign of the overall toroidal moment can be changed accordingly. Hence, the utilization of the electric toroidal moment appears unsuitable for distinguishing between CCW and CW polar spirals. The authors are advised to introduce an alternative parameter in place of the electric toroidal moment.

Ref: 23-42356-T

Title: Electrically and Mechanically Driven Rotation of Polar Spirals in a Relaxor Ferroelectric Polymer

Replies to Reviewers' Comments and Corresponding Changes

We sincerely thank the reviewers for their time and effort in carefully reviewing our manuscript. The comments and suggestions are very valuable, and led us to conduct further experiments and simulations. The manuscript has been revised accordingly. The revisions are highlighted in the manuscript. We also list the revisions and responses below.

Reviewer #1 (Remarks to the Author):

The manuscript by M. Guo et al. reported a non-volatile rotation of polar spirals in relaxor ferroelectric polymer P(VDF-TrFE) driven by either a small electric or mechanical field. This manuscript also discussed the formation of these polar spirals, arising from the asymmetric Coulomb interaction between vertically aligned helical polymer chains. These polar spirals also show robust retention and also support optical read out by AFM-IR.

I think this work presents a new possibility to manipulate the topological polar states in relaxor ferroelectric polymers. The results are important and intriguing, which can also inspire the field to explore similar phenomena in other inorganic materials. I would recommend publishing this paper in Nature Communications after addressing a few minor points:

Reply:

We thank the reviewer for his/her appreciation of the novelty and the broad impact of our work. We also appreciate the reviewer's valuable comments and suggestions on the manuscript, to which we respond below.

Comment 1:

I think the title of this paper can be slightly tailored since the paper only highlights the electrically driven behavior but the actual paper presents both the electrical and mechanical driven rotation.

Reply:

We thank the reviewer for the valuable suggestion. We have accordingly changed the title into 'Electrically and Mechanically Driven Rotation of Polar Spirals in a Relaxor Ferroelectric Polymer'.

Revisions: *The change to the title affects the manuscript, and the supplementary information.*

Comment 2:

I think it is helpful to comment or explain possible mechanisms that cause the differences in the effect of electrical and mechanical driven rotation. For example,

electrically induced rotation can reach 130 deg while the mechanically induced rotation can exceed 780 deg.

Reply:

We thank the reviewer for the insightful comment. The main difference between electrical and mechanical driven rotation is the range of the rotational angle. As shown in Fig. 2b, c, the rotational angle after each manipulation is positively associated with the magnitude of the external field. There should be two reasons why the rotation behaviors under electrical and mechanical field are different.

The first reason lies in the rotation mechanism. As indicated from the phase-field simulation on polar spirals, the electrical and mechanical driven rotation are related to the electro-mechanical coupling effects (Fig. 3f and Supplementary Fig. 19). For mechanical driven rotation, the stress-induced polarization rotation should be the primary effect, and thus the range of the rotational angle is wider. For electrical driven rotation, however, either upward or downward electric fields drive the rotation in the same sense. These suggest that the electrical rotation should be attributed to the electrostrictive effects which are coupled to the square of electric field, unlike the piezoelectric effects where upward and downward electric fields should have opposite effects. The electrostrictive stress should then further induce the rotation of the local polarization. Thus, the electrically driven rotation should be a secondary effect, that could lead to narrower range of the rotational angles than the mechanical manipulation. The second reason lies in the destruction mechanism. For mechanical manipulation, the destruction of the polar spirals is mainly induced by the plastic deformation of the polymer. While for electrical manipulation, the destruction of the polar spirals should be induced by both the plastic deformation from electrostrictive stress and large local electric field near the biased tip¹ that exceeds the switching coercive field. The synergistic effects of them should lead to early destruction of the polar spirals before further manipulation, leading to the narrower range of rotation when induced by electric fields. Therefore, the limit of the rotational angle in electrical manipulation is less than that in mechanical manipulation.

We revised the manuscript to discuss on the different rotation behavior in electrical and mechanical manipulation.

Revisions:

(1) Please see lines 4-7 (page 7) in the manuscript highlighted in red.

The rotational direction is the same when the bias takes a positive sign (Supplementary Fig. 8g-1 and Supplementary Fig. 9), suggesting that the electrical rotation should be attributed to the electrostrictive effects which are coupled to the square of electric field.

(2) Please see lines 17-21 (page 7) in the manuscript highlighted in red.

The narrower range of rotation induced by electric fields should be attributed to the indirect manipulation through electrostriction, and the early destruction of polar spirals induced by the plastic deformation and the large local electric field near the biased tip³⁰ that exceeds the switching coercive field.

Comment 3:

I would also like to see if the authors have conducted any macroscale electrical

measurements such as dielectric permittivity measurements for polar spirals having different rotation angles. If they indeed show different electrical response, these results can also enhance the paper and provide another electrical read out for the polar spirals.

Reply:

We thank the reviewer for the valuable suggestion. We have made efforts but found it difficult to obtain useful data from the macroscopic electrical measurements.

To measure the dielectric permittivity in the standard geometry of a parallel plate capacitor, we have tried to evaporate a layer of copper as the top electrode (diameter around 130 μm) onto a P(VDF-TrFE) film deposited on gold as the bottom electrode (Fig. R1a). However, when covered by copper (the region with greater height in Fig. R1b, on the right side of the yellow line), the domain structure of the polymer could no longer be observed, as shown by the uniform IP-PFM phase signals (Fig. R1c) and the weak IP-PFM amplitude signals (Fig. R1d). The unobservability of domain structures then disable us to locate and manipulate the polar spirals, which hamper their further measurements. However, we suggest studying them in the future and add it as an outlook in the revised manuscript.

Fig. R1. The ferroelectric domain structures near electrodes. **a** Optical image of copper electrodes evaporated on the P(VDF-TrFE) film. The diameter of electrodes is around 130 μm . The scale bar is 100 μm . **b-d** AFM morphology (**b**), IP-PFM phase (**c**) and IP-PFM amplitude (**d**) images near copper electrodes. Morphology with greater height refers to the electrode region. The P(VDF-TrFE) film and copper electrode regions are separated by the yellow lines in **b-d**. The scale bar is 3 μm .

Revisions: Please see lines 14-17 (page 13) in the manuscript highlighted in red.

The polar spirals can be rotated by applying electric fields or mechanical stresses, enabling non-destructive, non-volatile, and continuous rotations, which may have influence on the macroscopic dielectric properties and is promising for multistate memories and neuromorphic systems³⁴.

Comment 4:

It is a slight pity to see that the polar spirals can only rotate along one direction given their polar direction. Can the authors also comment whether it is possible to induce rotation for them to rotate along the another direction (for example, CW polar spiral rotates along CCW direction). In that way, it is possible to have a reversible switching to reset the polar spiral back to its initial state.

Reply:

We thank the reviewer for the valuable comment. We have made many attempts to reverse the sense of rotation. However, new experiment and simulation results suggest the polar spirals can only rotate in one sense under various field conditions.

Initially, since the polar spirals are in-plane textures, we explored the effect of the in-plane trailing field during AFM scanning on the sense of rotation. The trailing field is the equivalent in-plane field induced by electric² or flexoelectric³ field during the scanning process. It is parallel to the slow scan direction and has been shown to effectively control the in-plane polarization in some cases^{2,3}. Thus, we supposed the in-plane trailing field might be effective in inducing the rotation in different senses. However, when the slow scan axes were rotated, the CW/CCW polar spirals always show a rotation in the CW/CCW sense. These suggest that the rotation of the in-plane spirals is primarily induced by the out-of-plane components of the fields, regardless of their in-plane directions.

Fig. R2. The rotation behavior of polar spirals with different slow scan directions. **a** Schematic illustration of AFM scanning process. The trailing field is parallel to the slow scan axis, and is perpendicular to the fast scan axis. The four-headed-arrow defines slow scan directions in **b**. **b** The rotational angle of polar spirals with different slow scan directions. The CW (CCW) polar spiral rotating along the CW (CCW) direction, irrelevant to the slow scan directions. The insets in **b** show the IP-PFM images during the field manipulation. The side length of each inset is 2 μm .

It is more interesting to note that all the mechanical stress along out-of-plane directions should be compressive when the polar spirals are mechanically or electrically driven to

rotate. This is evident for the mechanical manipulations from the tip-based compressions. For the electrical manipulation, since the rotation should be induced through electrostriction, either upward or downward electric fields should generate compressive electrostriction strain onto the polymer. Therefore, it is quite consistent to find that all the electrical manipulation and the compressive mechanical manipulation to induce CW/CCW polar spirals to rotate in the CW/CCW sense.

This phenomenon then stimulated our interest in investigating the effect of a tensile stress along out-of-plane direction. Considering that this would be difficult to realize in experiments, we performed phase-field simulations and it shows that the CW/CCW polar spirals still rotate in the CW/CCW sense under the tensile stress (Fig. R3). This suggests that the sense of field driven rotation is irrelevant to the sign of out-of-plane strain.

Fig. R3. Phase-field simulations of polar spirals in a thin film under tensile stress. **a-c** Polarization maps within a CCW polar spiral in its initial state (**a**) and after the application of tensile stress of 2.64 MPa (**b**), 3.60 MPa (**c**). **d-f** Polarization maps within a CW polar spiral in its initial state (**d**) and after the application of tensile stress of 2.64 MPa (**e**), 3.60 MPa (**f**). The color of vectors depicts the divergence of local polarization.

Although it is hard to reset a polar spiral by a reversal in the rotation, we can in effect reset a polar spiral back to its initial state after rotating it by 360 deg (Fig. R4). Therefore, we have revised the manuscript to discuss the irrelevance of the sense in rotation to external field and the reset of a polar spiral by rotating 360 deg.

Fig. R4. The reset of polar spirals by rotating 360 degrees. **a-c** IP-PFM images of a CW polar spiral before any rotation (**a**), after CW rotation by 225° (**b**) and further CW rotation by 352° (**c**). The scale bar is 0.3 μm . **d-f** The divergence of local polarization in the same region as **a-c** before any rotation (**d**), after CW rotation by 225° (**e**) and further CW rotation by 352° (**f**). **g, h** The histograms of rotational angle and a fitting to the normal distribution yielding a Gaussian peak centered at 225° (state **a** to **b**) and 352° (state **a** to **c**), respectively.

Revisions: Please see lines 21-22 (page 7) and line 1 (page 8) in the manuscript and Supplementary Fig. 12 (page 15) in the supplementary information highlighted in red.

Although the sense of rotation is independent on the external field, we can in effect reset a polar spiral back to its initial state after rotating it by 360° (Supplementary Fig. 12).

Reviewer #2 (Remarks to the Author):

This work reports the non-volatile rotation of polar spirals in discretized microregions of poly(vinylidene fluoride-trifluoroethylene) random copolymer caused by small electric or mechanical fields. The results, which are well supported by abundant experimental evidences, are quite interesting and may of great importance for the application of complex materials. The methods used in this work are detail and reproducible. It meets the expected standards in this field and can thus be accepted for publication in Nature Communication essentially as is or with some minor revision.

Reply:

We sincerely thank the reviewer for his/her appreciation of our work, by recognizing our work as “well supported by abundant experimental evidences”, “quite interesting” and “of great importance”. We would also like to thank the reviewer for his/her valuable comments and suggestions on the manuscript. We will respond to these comments one by one below. The response and revisions are highlighted in red.

Comment 1:

As can be seen from Fig. 1a, there are cracks in the film. I wonder whether the cracks influence the measurement or not. How about the roughness of the used film?

Reply:

We thank the reviewer for the comment. We note that these cracks have not influenced the measurements on the polar spirals and the films are smooth in surface:

- Big cracks with greater height undulation are crystal boundaries (as seen in Supplementary Fig. 3b, c, and region in Fig. R5 denoted by green squares), while our observations and manipulations have been confined to single crystals (Supplementary Fig. 14b, and regions in Fig. R5 denoted by yellow squares).
- Small cracks with smaller height undulation are the wrinkles within the single crystals (denoted by cyan squares in Fig. R5). These cracks only appear in region with curly stripe domains, and therefore have not influenced our experiments performed within the microregions exhibiting polar spirals. The cracks would be reduced in depth and length when rising temperature (denoted by yellow squares in Fig. R6), with the surface roughness R_a declined from 2.061 nm at 25°C (Fig. R6a) to 1.033 nm at 80°C (Fig. R6b).

Fig. R5. The domain structures with different morphologies. a, b AFM morphology (a) and IP-PFM phase (b) of a region in a relaxor ferroelectric polymer thin film. The green squares refer to the big crack regions (crystal boundaries), exhibiting stripe

domains. The cyan squares refer to the small crack regions (wrinkles), exhibiting stripe domains. The yellow squares refer to the polar spiral regions without cracks, exhibiting concentric ring-shaped domains. The scale bar is 1 μm .

Fig. R6. Thermal evolution of the morphology of a relaxor ferroelectric thin film. **a, b** AFM morphology of P(VDF-TrFE) face-on lamellae at 25°C (**a**) and 80°C (**b**). The yellow squares denote the small crack regions which would be reduced in depth and length when rising temperature. The scale bar is 0.4 μm .

We also evaluated the roughness from the AFM morphology images of the used film. The surface roughness R_a of the region in Supplementary Fig. 3b, c is 3.335 nm, 3.677 nm, respectively. The surface roughness of polar spirals is much lower, with $R_a = 1.568$ nm for Supplementary Fig. 3j. Thus, we suggest the surface roughness is relatively too small to influence the measurement.

We have added the discussion about the smooth surface of polar spiral regions.

Revisions: Please see lines 3-4 (page 17) in the manuscript highlighted in red.

The polar spiral regions we characterized and manipulated are smooth in surface without cracks (surface roughness $R_a = 1.568$ nm for Supplementary Fig. 3j).

Comment 2:

Moreover, the thickness of the film used in this work is about 100 nm. Considering that the lamellar thickness of the copolymer is generally around 10 nm, does the sample contain a monolayer lamella? If not, does the folding loops between the lamellae affect the related properties?

Reply:

We appreciate the reviewer's careful reading and for commenting on the thickness of the lamellar. Polymer chains can fold back and forth to form lamella crystals. The lamellar thickness varies from several to hundreds of nanometers, depending on the chemical composition of polymers and the crystallization dynamics. Table R1 shows the typical lamellar thickness of different polymers when they are crystallized from the melt or annealed at temperature near melting point. For PVDF-based polymers, the typical lamellar thickness is 100~200 nm.

Table R1. Typical lamellar thickness of polymers

Polymer	Lamellar thickness (nm)	References
Poly(ethylene terephthalate)	4~9	Ref. ⁴
Poly(L-lactide)	4~24	Ref. ⁵
High-density polyethylene	10~40	Ref. ⁶
β -polypropylene	20~60	Ref. ⁷
PVDF	100~200	Ref. ⁸
P(VDF-TrFE)	~100	Ref. ⁹
(72/28, 75/25, 81/19 mol%)		
P(VDF-TrFE)	100~200	Ref. ^{10, 11}
(70/30 mol%)		
P(VDF-TrFE)	~100	This work
(50/50 mol%)		

Since the lamellar thickness also depends on the crystallization dynamics, different crystallization processes might generate varied lamellar thickness of the same polymer. We noticed that the P(VDF-TrFE) copolymer with a different molar ratio has been reported to exhibit lamellar thickness of 25 nm after mechanically driven crystallization at 135°C below the melting point (150°C)¹². This is different to our melt-recrystallization process that can fully stretch the polymer chains to increase the chain fold period and thus obtain thicker lamellae around 100 nm¹³. The lamellae thickness of ~100 nm is verified in thicker films of the copolymer (Fig. R7), which are prepared by the same method as the 100-nm-thick film. Therefore, the 100-nm-thick sample in this study contains a monolayer lamella. To avoid the misunderstanding on the lamellar thickness, a relevant description has been added in the revised manuscript.

Fig. R7. Cross sectional SEM image of multilayer face-on lamellae relaxor ferroelectric thin film. The multilayer face-on lamellae film is also fabricated by spin-coating and melt-recrystallization methods. The typical lamellar thickness of relaxor ferroelectric thin film is around 100 nm. The scale bar is 1 μm.

Revisions: Please see lines 4-7 (page 4) in the manuscript highlighted in red.

By melt-recrystallizing spin-coated thin films of P(VDF-TrFE) (see dielectric properties in Supplementary Figs. 1, 2), a 100-nm-thick monolayer of face-on lamellae with vertically aligned polymer chains were produced (Fig. 1a and Supplementary Fig. 3a-c).

Comment 3:

In Fig. 2, the CW and CCW rotations are quite clear. However, those in Fig. 1g and h seem not so clear.

Reply:

Thanks for the comment. Fig. 2 depicts the rotations of CW/CCW polar spirals. However, Fig. 1g and h are the curl (g) and the divergence (h) of a static CCW polar spiral without rotations. We realize misunderstanding might be caused and thus revise relevant discussions to emphasize the content of those figures.

Revisions: Please see lines 6-10 (page 5) in the manuscript highlighted in red.

By assuming a finite calculus limit, we calculated the curl ($\nabla \times \mathbf{P}$, Fig. 1g) and divergence ($\nabla \cdot \mathbf{P}$, Fig. 1h) of the local polarization \mathbf{P} (Supplementary Fig. 4f), both of which present a double spiral that fills nearly all of the microregion. The positive/negative curl (red/blue spiral in Fig. 1g) denotes a counterclockwise/clockwise (CCW/CW) polarization rotation.

Comment 4:

The scale bar presented in Fig. 1e, 3b is not clear enough.

Reply:

Thanks for the good suggestion. We have deleted the texts of the scale bars in Fig. 1e and Fig. 3b and added description in the caption to avoid unclearness. Please see Fig. 1 and Fig. 3 in the revised manuscript.

Revisions: Please see Fig. 1 (page 30) and Fig. 3 (page 32) in the revised manuscript.

Reviewer #3 (Remarks to the Author):

Please see the attached file.

In this work, the authors presented an interesting study on the field-driven manipulation of the non-volatile rotation of polar spirals in discretized microregions of the relaxor ferroelectric polymer. The authors demonstrated that the polar spirals arise from the asymmetric Coulomb interaction between vertically aligned helical polymer chains and can be rotated in-plane through various angles with robust retention. There are, however, a few concerns and I'd like to see them addressed before the final consideration of publication.

Reply:

We greatly thank the reviewer for his/her appreciation of this work. We also thank the reviewer for the constructive comments and suggestions on the manuscript. We will respond the comments one by one below. The responses and revisions are highlighted in red.

Comment 4 (treated first as relevant for Comment 1):

There is controversy surrounding the use of the electric toroidal moment in Fig. S4. As depicted in Fig. S4a and c, both CCW and CW polar spirals exhibit a distribution of the electric toroidal moment comprising both local negative and positive components. However, when these local components are summed across the entire system, their contributions nearly cancel out, resulting in a value close to zero. In this study, if I understand correctly, the authors determine the overall toroidal moment by performing path integrals. However, it is important to note that the overall toroidal moment is heavily influenced by the selected integration path. By altering the chosen paths, the magnitude and sign of the overall toroidal moment can be changed accordingly. Hence, the utilization of the electric toroidal moment appears unsuitable for distinguishing between CCW and CW polar spirals. The authors are advised to introduce an alternative parameter in place of the electric toroidal moment.

Reply:

We thank the reviewer for the valuable comments. We would like to first clarify that we defined CCW and CW polar spirals by the sense of rotation that polar sink spirals outwards, instead of by referring to the sign of the electric toroidal moments. We would also like to clarify that the electric toroidal moments calculated in the original Fig. S4 are not path integrals but area integrals. The areas are circles centered at the spirals, and therefore should clearly demonstrate the non-zero electric toroidal moment in the polar spiral. And although they exhibit both positive and negative values, the CW/CCW polar spirals weigh more on the positive/negative than the other sign (Fig. R8 and Supplementary Fig. 7).

However, we realize that the mere use of area integrals may cause misunderstanding. We therefore plotted both path- and area-dependent electric toroidal moment versus the radius of the circular paths or areas to eliminate the potential misunderstanding (Fig. R8 and Supplementary Fig. 7), which similarly shows that the oscillating electric toroidal moment weighs more on either positive or negative side, and the electric toroidal moment of CW/CCW spirals are opposite. We have also added the discussion

about the path-dependent and area-dependent electric toroidal moment of polar spirals in supplementary information.

Fig. R8. Electric toroidal moment of CCW and CW polar spirals. **a** Distribution of the electric toroidal moment of a CCW polar spiral. The yellow star denotes its geometric center point. The concentric black circles illustrate the circular rings where the electric toroidal moments path integrated, as well as the circular areas where the electric toroidal moments area integrated. The scale bar is $0.3 \mu\text{m}$. **b, c** Profile of the path-dependent (**b**) and area-dependent (**c**) electric toroidal moment G_z versus radius of the integrated circle. The grey dashed line denotes the line of $G_z=0$. **d** Distribution of the electric toroidal moment of a CW polar spiral. The yellow star denotes its geometric center point. The concentric black circles illustrate the circular rings where the electric toroidal moments path integrated, as well as the circular areas where the electric toroidal moments area integrated. **e, f** Profile of the path-dependent (**e**) and area-dependent (**f**) electric toroidal moment G_z versus radius of the integrated circle. The grey dashed line denotes the line of $G_z=0$. The value of local polarization is adopted from the spontaneous polarization deduced from the electric hysteresis of the relaxor ferroelectric polymer in Supplementary Fig. 2.

Revisions: Please see lines 1-11 (page 9) and Supplementary Fig. 7 (pages 9-10) in the revised supplementary information highlighted in red.

In this study, the electric toroidal moment is calculated to describe the polarization rotation behavior of polar spirals. To comprehensively illustrate this behavior, we calculate both the path-dependent () and area-dependent

() toroidal moment^{S1}, as shown in Fig. S7. The integral path and area are a circular ring and circle, respectively, and are centered at the spirals. The path-dependent and area-dependent toroidal moments exhibit similar behaviors. Their values

oscillate as integral radius increases, and weigh more on the positive/negative than the other sign for the CW/CCW polar spirals. Since the integral area covers major region of polar spirals, we confirm the emergence of toroidal order in the polar spirals based on its non-zero electric toroidal moment, where CW polar spirals exhibit electric toroidal moments with positive values, and CCW ones exhibit negative values.

Comment 1:

The reasons why CW spirals are more pronounced in the considered system compared to CCW counterparts remain unclear. Given that this study primarily investigates the control of polar spirals, it is essential to elucidate the underlying mechanism responsible for their formation.

Reply:

We thank the reviewer for the good question. First, we note that CCW spirals are more pronounced, as the CW/CCW ratio is 0.85.

The sense of spirals is defined based on the 2-dimensional case from the top view (Fig. R9a). According to electric toroidal moment we calculated (Fig. R8), the CW/CCW spirals have opposite electric toroidal moment. This suggests that CW/CCW spirals might be equivalent in 3-dimensions, and can be converted to each other by a 180° rotation along any in-plane axis (Fig. R9a-d). The equivalence of CW and CCW spirals suggests that the polar spirals we observed (Fig. R9c, d) are monochiral, and their topological enantiomers (Fig. R9e, f) have not been observed in our study. Since the P(VDF-TrFE) face-on lamellae have out-of-plane polarization¹¹, there can theoretically be an out-of-plane component of the polar spirals, although we have not clearly observed it, which might be attributed to the electrostatic screening and its small magnitude. But due to the dielectric asymmetry between the air and the substrate along out-of-plane direction, the ratio of the out-of-plane alignment should be uneven. Similar phenomena that exhibit preferentially aligned dipoles in out-of-plane directions in ferroelectric films have been reported and ascribed to asymmetric effects, e.g., compositional and strain gradient^{14,15}, pinning defect dipoles¹⁶, asymmetric electrodes¹⁷, charge trapping at interfaces¹⁸, surface adsorbates¹⁹, etc.

However, it is hard to characterize the 3-dimensional polarization distribution of polar spirals, as there is still no technique to obtain the cross-sectional information of polymer thin films. Therefore, we prefer to keep the current framework which is neat and uncontroversial. But relevant discussion about the possible reason why CW and CCW spirals are different in quantity has been added in the manuscript.

Fig. R9. The relationship between CW/CCW polar spirals. **a, b** Schematic illustration of asymmetric air/film/substrate trilayer system with definition from top view (**a**) and bottom view (**b**). **c-d** The CW ($+G_z$, **c**) and CCW ($-G_z$, **d**) polar spirals that have been observed in this study. The CW/CCW spirals defined from top view would be the same as CCW/CW spirals defined from bottom view, possessing a 180° rotational relationship along horizontal axis with respect to each other. **e-f** The CCW ($+G_z$, **e**) and CW ($-G_z$, **f**) polar spirals that have not been observed in this study. The **e** and **f** are the topological enantiomers of **c** and **d**, respectively.

Revisions: Please see lines 4-9 (page 6) in the manuscript highlighted in red.

It should be noted that CW/CCW polar spirals possess a positive/negative electric toroidal moment (Supplementary Fig. 7), indicating that the CW and CCW polar spirals might be mono-chiral and can be converted to each other by a 180° rotation along any in-plane axis. Thus, the uneven distribution might be attributed to the dielectric asymmetry between the air and the substrate.

Comment 2:

As shown in Fig. 2, the application of either electric field or mechanical stress solely alters the rotation angle of polar spirals without affecting their orientation, whether it is CW or CCW. Is it feasible to modify the rotation of polar spirals, switching them from CCW to CW or vice versa, using external fields? It is worth noting that achieving complete control over polar topological textures necessitates the ability to manage both their formation and switching among the equivalent states.

Reply:

Thanks for the insightful comment. The sense of rotation can only be changed when the microregion has been transformed into phases with higher symmetry and then transformed back. Therefore, the sense of CCW or CW is protected against high fields whilst the manipulation.

We further examined the thermal evolution of polar spirals and discovered that the sense of polar spirals could be protected after thermal treatment in paraelectric phase. The initial region mainly contains three CW polar spirals (Fig. R9a-c, denoted by the yellow squares). After annealed at 140°C (paraelectric phase), the polar spirals remained in

almost the same region, and the CW rotation was protected (Fig. R9d-f, denoted by the yellow squares). It should be attributed to the same 3/1 helical conformation²⁰ (Supplementary Fig. 15a) in both the relaxor ferroelectric phase and the paraelectric phase²¹. After recrystallization at 170 °C (molten phase), CCW polar spirals that had not appeared before thermal treatment were induced (Fig. R9g-i, denoted by the cyan squares). This suggests that the sense of rotation could only be changed when polar spirals are transformed into phases with higher symmetry and then transformed back. However, electric or mechanical fields can only generate irreversible transitions into monodomains or curly stripe domains, indicating that the protection of a spiral is so strong that it can only be destroyed by these fields. The discussion about the protection of the spiral sense has been added in the revised manuscript.

Fig. R9. Thermal evolution of polar spirals. **a-c** AFM morphology (**a**) and angle-resolved IP-PFM phase (**b, c**) images before any thermal treatment. The sample rotational angles in **b, c** are 0°, 45°, respectively. Sample rotation along CW direction, which is the measurement axis rotation along CCW direction, is defined with a positive sign. The arrows at the upper right of each figure denote the measurement axes. **d-f** AFM morphology (**d**) and angle-resolved IP-PFM phase (**e, f**) images after annealed at 140 °C for 10 min. The sample rotational angles in **e, f** are 0°, 45°, respectively. **g-i** AFM morphology (**g**) and angle-resolved IP-PFM phase (**h, i**) images after annealed at 170 °C for 10 min. The sample rotational angles in **h, i** are 0°, 45°, respectively. The yellow (cyan) squares denote CW (CCW) polar spirals. The scale bar is 2 μm.

Revisions: Please see lines 1-2 (page 8) in the manuscript highlighted in red.

Besides, the sense of polar spirals (CW or CCW) is strongly protected against the electric or mechanical field whilst manipulation.

Comment 3:

In the phase field model, the authors described the elastic energy density as

where, there exist three independent elastic constants c_{11} , c_{12} and c_{44} in the Voigt's notation. However, this description of elastic energy density assumes linear elasticity and cubic symmetry, which is not suitable for ferroelectric polymer. Ferroelectric polymers typically exhibit nonlinear behavior and lack cubic symmetry. Therefore, the authors should adjust the phase field model to appropriately account for the characteristics of ferroelectric polymer. Additionally, it is important to note that the elastic energy density has a significant contribution to the total energy of the considered system, as shown in Fig. 3f. This raises a critical concern about the validation of phase field simulations due to the present assumption.

Reply:

We thank the reviewer for this helpful comment. We would like to note that the non-cubic symmetry has been considered in terms of eigenstrain, and our phase-field simulations mainly consider the linear mechanical behavior (elastic) region.

The description of the elastic energy density in phase-field model usually adopts the usage of elastic constants in the parent cubic phase and includes the effect of different lattice symmetry in the term of eigenstrain. The eigenstrain is proportional to the square of polarization, which contains the spontaneous polarization induced by the symmetry breaking from the parent cubic phase. Therefore, we would like to suggest that the current phase-field framework has already considered the non-cubic symmetry of the ferroelectric polymer. We would also like to note that this method has been widely applied and verified to be suitable in oxide systems such as BaTiO_3 (tetragonal symmetry^{22,23}), BiFeO_3 (rhombohedral and tetragonal symmetry²⁴⁻²⁶), and $\text{Pb}(\text{Zr},\text{Ti})\text{O}_3$ (tetragonal symmetry²⁷).

As for the issue of the nonlinear mechanical behavior, we would like to suggest that the phase-field parameters during the nonlinear plastic deformation is quite difficult to determine and therefore we only consider the linear elastic region. For instance, the strains applied in simulation do not exceed 12.0 MPa, which are more than 2 to 3 orders of magnitude lower than the assumed elastic constant ($c_{11} = 48800 \text{ MJ m}^{-3}$, $c_{12} = 5600 \text{ MJ m}^{-3}$, $c_{44} = 21600 \text{ MJ m}^{-3}$, and $1 \text{ MJ m}^{-3} = 1 \text{ MPa}$). This suggests that the strain is too small to induce a nonlinear mechanical behavior, e.g., plastic deformation. However, we consider the phase-field simulation to be effective for semi-qualitatively addressing the rotational mechanism of the polar spirals at low field before destruction, i.e., in the elastic region. We have revised the manuscript to discuss the limitation of our phase-field simulations.

Revisions: Please see lines 8-11 (page 20) in the manuscript highlighted in red.

In this study, only the linear elastic region is considered when simulating the low-field response of the polar spirals. Thus, the strains applied in simulation do not exceed 12.0 MPa, which is lower than the assumed elastic constant by 2-3 orders of magnitude (Supplementary Table 3).

References

1. Soergel, E. Piezoresponse force microscopy (PFM). *Journal of Physics D: Applied Physics* **44** (2011).
2. Crassous, A., Sluka, T., Tagantsev, A. K. & Setter, N. Polarization charge as a reconfigurable quasi-dopant in ferroelectric thin films. *Nature Nanotechnology* **10**, 614-618 (2015).
3. Park, S. M. et al. Selective control of multiple ferroelectric switching pathways using a trailing flexoelectric field. *Nature Nanotechnology* **13**, 366-370 (2018).
4. Xia, Z. Y., Sue, H. J., Wang, Z. G., Avila-Orta, C. A. & Hsiao, B. S. Determination of crystalline lamellar thickness in poly(ethylene terephthalate) using small-angle X-ray scattering and transmission electron microscopy. *Journal of Macromolecular Science-Physics* **B40**, 625-638 (2001).
5. Kanchanasopa, M., Manias, E. & Runt, J. Solid-State Microstructure of Poly(L-lactide) and L-Lactide/meso-Lactide Random Copolymers by Atomic Force Microscopy (AFM). *Biomacromolecules* **4**, 1203-1213 (2003).
6. Zhou, H. Y. & Wilkes, G. L. Comparison of lamellar thickness and its distribution determined from dsc, SAXS, TEM and AFM for high-density polyethylene films having a stacked lamellar morphology. *Polymer* **38**, 5735-5747 (1997).
7. Trifonova, D., Varga, J. & Vancso, G. J. AFM study of lamellar thickness distributions in high temperature melt-crystallization of β -polypropylene. *Polymer Bulletin* **41**, 341-348 (1998).
8. Hattori, T., Kanaoka, M. & Ohigashi, H. Improved piezoelectricity in thick lamellar β - form crystals of poly(vinylidene fluoride) crystallized under high pressure. *Journal of Applied Physics* **79**, 2016-2022 (1996).
9. Ohigashi, H., Akama, S. & Koga, K. Lamellar and Bulk Single Crystals Grown in Annealed Films of Vinylidene Fluoride and Trifluoroethylene Copolymers. *Japanese Journal of Applied Physics* **27**, 2144-2150 (1988).
10. Guo, M. et al. Flexible Robust and High-Density FeRAM from Array of Organic Ferroelectric Nano-Lamellae by Self-Assembly. *Adv Sci* **6**, 1801931 (2019).
11. Guo, M. et al. Toroidal polar topology in strained ferroelectric polymer. *Science* **371**, 1050 (2021).
12. Jung, H. J. et al. Shear-Induced Ordering of Ferroelectric Crystals in Spin-Coated Thin Poly(vinylidene fluoride-co-trifluoroethylene) Films. *Macromolecules* **42**, 4148-4154 (2009).
13. Hikosaka, M., Sakurai, K. & Koizumi, H. O. Morphology of Extended Chain Single Crystals of Vinylidene Fluoride and Trifluoroethylene Copolymers. *Japanese Journal of Applied Physics* **32**, 2029 (1993).
14. Mangalam, R. V. K., Karthik, J., Damodaran, A. R., Agar, J. C. & Martin, L. W.

- Unexpected Crystal and Domain Structures and Properties in Compositionally Graded $\text{PbZr}_{1-x}\text{Ti}_x\text{O}_3$ Thin Films. *Advanced Materials* **25**, 1761-1767 (2013).
15. Agar, J. C. et al. Highly mobile ferroelastic domain walls in compositionally graded ferroelectric thin films. *Nature Materials* **15**, 549-556 (2016).
 16. Damodaran, A. R., Breckenfeld, E., Chen, Z., Lee, S. & Martin, L. W. Enhancement of Ferroelectric Curie Temperature in BaTiO_3 Films via Strain-Induced Defect Dipole Alignment. *Advanced Materials* **26**, 6341-6347 (2014).
 17. Lee, J., Choi, C. H., Park, B. H., Noh, T. W. & Lee, J. K. Built-in voltages and asymmetric polarization switching in $\text{Pb}(\text{Zr},\text{Ti})\text{O}_3$ thin film capacitors. *Applied Physics Letters* **72**, 3380-3382 (1998).
 18. Lee, E. G., Wouters, D. J., Willems, G. & Maes, H. E. Voltage shift and deformation in the hysteresis loop of $\text{Pb}(\text{Zr},\text{Ti})\text{O}_3$ thin film by defects. *Applied Physics Letters* **69**, 1223-1225 (1996).
 19. Lee, H. et al. Imprint Control of BaTiO_3 Thin Films via Chemically Induced Surface Polarization Pinning. *Nano Letters* **16**, 2400-2406 (2016).
 20. Liu, Y. et al. Ferroelectric polymers exhibiting behaviour reminiscent of a morphotropic phase boundary. *Nature* **562**, 96-100 (2018).
 21. Liu, Y. et al. Chirality-induced relaxor properties in ferroelectric polymers. *Nat Mater* **19**, 1169-1174 (2020).
 22. Guo, C. Q. et al. Domain evolution in bended freestanding BaTiO_3 ultrathin films: A phase-field simulation. *Applied Physics Letters* **116** (2020).
 23. Zhou, Y. Q. et al. Tip-Induced In-Plane Ferroelectric Superstructure in Zigzag-Wrinkled BaTiO_3 Thin Films. *Nano Letters* **22**, 2859-2866 (2022).
 24. Peng, R.-C. et al. Boundary conditions manipulation of polar vortex domains in BiFeO_3 membranes via phase-field simulations. *Journal of Physics D: Applied Physics* **54**, 495301 (2021).
 25. Peng, R. C. et al. Domain patterns and super-elasticity of freestanding BiFeO_3 membranes via phase-field simulations. *Acta Materialia* **208** (2021).
 26. Zhang, Y. Y. et al. Controlled Nucleation and Stabilization of Ferroelectric Domain Wall Patterns in Epitaxial (110) Bismuth Ferrite Heterostructures. *Advanced Functional Materials* **30** (2020).
 27. Indergand, R., Vidyasagar, A., Nadkarni, N. & Kochmann, D. M. A phase-field approach to studying the temperature-dependent ferroelectric response of bulk polycrystalline PZT. *Journal of the Mechanics and Physics of Solids* **144** (2020).

REVIEWERS' COMMENTS

Reviewer #1 (Remarks to the Author):

The authors have answered all my questions satisfactorily and have made corresponding revisions in the manuscript. I am glad to recommend publishing this paper in Nature Communications.

Reviewer #2 (Remarks to the Author):

Please see the attachment

Reviewer #3 (Remarks to the Author):

The authors have addressed all my comments in details. However, there is a small mistake in their response and revised manuscript. Specifically, the author state at lines 409 and 410 in the revised manuscript that "the strains applied in simulation do not exceed 12.0 MPa". The term "strains" must be replaced by "stress".

Generally, I would like to recommend for publication in Nature Communications.

As mentioned in my previous comments, this work reports some interesting results about the non-volatile rotation of polar spirals in discretized microregions of poly(vinylidene fluoride-trifluoroethylene) random copolymer caused by small electric or mechanical fields, which are of great importance for the application of complex materials. The authors have revised the manuscript with due consideration of my comments. In this case, I would like to support the acceptance of this work for publishing on Nature Communication.

Ref: 23-42356A

Title: Electrically and Mechanically Driven Rotation of Polar Spirals in a Relaxor Ferroelectric Polymer

Replies to Reviewers' Comments and Corresponding Changes

We sincerely thank the reviewers for their time and effort in carefully reviewing our manuscript once again, as well as their recommendations for publication in *Nature Communications*. We also appreciate the comments from reviewers, and the manuscript has been revised accordingly. The point-by-point response and revisions are listed below.

Reviewer #1 (Remarks to the Author):

The authors have answered all my questions satisfactorily and have made corresponding revisions in the manuscript. I am glad to recommend publishing this paper in Nature Communications.

Reply:

We greatly appreciate the positive feedback from the reviewer regarding our responses and revised manuscript, as well as his/her recommendation.

Reviewer #2 (Remarks to the Author):

As mentioned in my previous comments, this work reports some interesting results about the non-volatile rotation of polar spirals in discretized microregions of poly(vinylidene fluoride-trifluoroethylene) random copolymer caused by small electric or mechanical fields, which are of great importance for the application of complex materials. The authors have revised the manuscript with due consideration of my comments. In this case, I would like to support the acceptance of this work for publishing on Nature Communication.

Reply:

We sincerely appreciate the reviewer for his/her recognition of the broad impact of our work, as well as his/her recommendation.

Reviewer #3 (Remarks to the Author):

The authors have addressed all my comments in details. However, there is a small mistake in their response and revised manuscript. Specifically, the author state at lines 409 and 410 in the revised manuscript that "the strains applied in simulation do not exceed 12.0 MPa". The term "strains" must be replaced by "stress".

Generally, I would like to recommend for publication in Nature Communications.

Reply:

We greatly thank the reviewer for his/her appreciation and recommendation. We also thank the reviewer for pointing out the mistakes in our responses and revised manuscript. The corresponding revisions are highlighted in red.

Revisions: Please see lines 9-11 (page 20) in the manuscript highlighted in red.

Thus, the stresses applied in simulation do not exceed 12.0 MPa, which is lower than the assumed elastic constant by 2-3 orders of magnitude (Supplementary Table 3).